# OpenReview forum: "Narrow Finetuning Leaves Clearly Readable Traces in Activation Differences"
_ICLR.cc/2026/Conference — ICLR 2026 Poster_

### Official Review · Reviewer_Bkbd · 2025-10-29

**Soundness:** 4
**Presentation:** 4
**Contribution:** 3
**Rating:** 8
**Confidence:** 3

**Summary:**

The paper introduces ADL method which uses model diffing techniques to investigate traces left by narrow finetuning. The traces are visible even for unrelated inputs and the authors identify, reason and provide strategies to mitigate this effect.

**Strengths:**

* Thorough experimental setup and extensive results
* Novel methodology that effectively leverages existing techniques to identify traces
* The analysis provides very important inferences which will be beneficial for the community about safety in narrow fine-tuned models
* It also provides suggestions on how to mitigate this effect

**Weaknesses:**

The causal effect is positive for Gemma3 which contradicts the hypothesis and warrants more investigation

**Questions:**

* The authors specify in line 161 that they focus on the middle layer. Could they expand on why?
* Edit suggestion: Explicit mention of the method name (ADL) when introducing the approach in the abstract would be good

---

> ### Author Response · Authors · 2025-11-21
>
> We thank the reviewer for their very positive feedback and constructive input! We now respond to the points in detail.
>
> **Weaknesses**:
> > The causal effect is positive for Gemma3 which contradicts the hypothesis and warrants more investigation
>
> We would like to highlight that the results do actually hold up for gemma: On the finetuning dataset the intervention results in a spike in loss. The only thing that is inconsistent with gemma is that the loss also increases on the pretraining dataset, which we hypothesis is because “this model changed sufficiently during finetuning that the ablated directions became crucial for the finetuned model's computation, making replacement with base model activations generally harmful.” (L456). While we do agree that it might be interesting to verify our hypothesis, we believe it is beyond the scope of the paper and does not influence the main takeaway.
>
>
> **Questions**:
>
> > The authors specify in line 161 that they focus on the middle layer. Could they expand on why?
>
> This choice builds on a range of literature that shows that middle layers are where we expect to find the richest and most interesting representations [1,2] (we have added a reference to those). There’s a range of interpretability literature that has focused on the middle literature because of this (e.g. [2,3]).
>
> > Edit suggestion: Explicit mention of the method name (ADL) when introducing the approach in the abstract would be good
>
> Good point, we have implemented this.
>
> [1]: Skeal et al.: Layer by Layer: Uncovering Hidden Representations in Language Models
>
> [2]: Ali et al.: Entropy-Lens: The Information Signature of Transformer Computations
>
> [2]: Lindsey et al.: Sparse Crosscoders for Cross-Layer Features and Model Diffing
>
> [3]: Minder et al.: Overcoming Sparsity Artifacts in Crosscoders to Interpret Chat-Tuning

---

### Official Review · Reviewer_jbUP · 2025-10-30

**Soundness:** 3
**Presentation:** 4
**Contribution:** 3
**Rating:** 8
**Confidence:** 3

**Summary:**

The work uncovers biases in narrowly fine-tuning models' activation spaces and hypothesizes the origins of said biases, and provides a possible direction to mitigate such biases.

**Strengths:**

1) I think the paper is written very well.
2) I think the goal and contribution of work are very relevant to interpretability research as a whole.
3) The experiments are sound and aid the central claims of the work.
4) The finding is very interesting and well-grounded in current literature. Although the paper could use some more explanations of the methods utilized in the work, such as PathScope. I encourage the authors to add further explanations about this methodology in the final iterations of the work.
5) Mixing pre-training data with the fine-tuning one is an interesting approach; I would like some more emphasis on this in the final iteration of the paper.
6) Overall, the work aims to present and discuss key facts about fine-tuning for narrow use cases, particularly the data dependence of bias in fine-tuned models.

**Weaknesses:**

The major weakness of note is that the bias mitigation method proposed lacks a study of the impact of injecting pre-training data into the fine-tuning object. Both in the sense of measuring the domain-specific capabilities impacted by the model, and the added cost of adding more data to the fine-tuning objective. Although the work does claim that adding pre-training data reduces fact alignment scores, a more rigorous analysis would be appreciated.

Another weakness, more so of a comment, to highlight would be that the methodology presented is just a collection of prior known/used methods, although this doesn't undermine the novelty of the findings; highlighting the derivative nature of the methodology would help elucidate the contributions of the work.

Even though, as mentioned previously, i think the paper is written very well, continuous references to the appendix break the flow of the paper. I would encourage that in the final iteration of the work, to add these appendix section to the main paper or leave some as a footnote.

**Questions:**

Would the results be consistent across multiple graders, it would be nice to know the amount of noise that could be mitigated if we use multiple graders.

---

> ### Author Response · Authors · 2025-11-21
>
> We thank the reviewer for their very positive feedback and constructive input! We now respond to the points in detail.
>
> **Weaknesses**:
> > Mixing in data has an impact on the finetuning objective.
>
> We’d like to highlight that this is precisely what the False Fact Alignment (FFA) score captures. Moreover, we believe that mixing in other pretraining data reflects a realistic scenario for SDF finetuning, as it simulates how such facts would have been learned during pretraining. The FFA  score is the mean of three metrics that measure the degree of false fact belief. These metrics are borrowed from [1]:
>
> - MCQ Distinguish: A multiple choice question with two options: one aligning with the true belief and one with the false belief.
> - Open-Ended Belief: An open-ended question about the inserted fact. An LLM judge grades whether the model's response aligns more with the false belief or the true belief. If the response is ambiguous, that data point is discarded.
> - Context Comparison: Both true and false universe contexts are presented to the model, and the model is asked to reason about which phenomenon is more likely to be true.
>
> We believe that this range of evaluations already proposes quite an extensive evaluation of whether the inserted facts are well represented by the model. We agree that we have not discussed the additional training cost but believe that this is fairly low as training these narrow finetunes are usually generally not really computationally that expensive and the mixing of data we investigated is at most a 2x increase (since it just grows linearly with the number of datapoints).
>
>
> > Another weakness, more so of a comment, to highlight would be that the methodology presented is just a collection of prior known/used methods, although this doesn't undermine the novelty of the findings; highlighting the derivative nature of the methodology would help elucidate the contributions of the work.
>
> We fully agree with this but believe we have highlighted this quite well. For example, we don’t mention ADL in the contributions and state “using Patchscope and steering techniques” or “we leverage Patchscope”. Nonetheless, we have added a clearer statement in the introduction “Our method, Activation Difference Lens (\ADL), \change{leverages two well established interpretability techniques.”
>
>
> > Even though, as mentioned previously, i think the paper is written very well, continuous references to the appendix break the flow of the paper. I would encourage that in the final iteration of the work, to add these appendix section to the main paper or leave some as a footnote.
>
> This is a good point. We have tried to improve this.
>
> **Questions**:
> > Would the results be consistent across multiple graders, it would be nice to know the amount of noise that could be mitigated if we use multiple graders.
>
> We agree with the reviewer and have added extensive analysis with respect to the effect of different graders in Section 4.1 as well as Appendix D. This includes:
> - Running the full agent also with Gemini 2.5 Pro additionally to GPT-5 (medium)
> - Extending the hypothesis grading to graders Gemini 2.5 Flash and Claude Haiku 4.5 including 3 reruns per agent hypothesis and grader model.
> - Running the token relevance grading and patchscope scaling factor grading additionally with Gemini 2.5 Lite and Claude Haiku 4.5.
> - Running the steering factor additionally with Gemini 2.5 Flash Lite
>
> We evaluate a) whether these graders result in systematic biases using hierarchical generalized linear models (GLM) within the HiBayes framework (https://arxiv.org/pdf/2505.05602, https://arxiv.org/html/2507.03772v2) and b) whether the graders agree through correlation analysis and Krippendorff’s alpha. Results show that the agent model has no systematic impact on the grade and all hypothesis graders agree to a very high degree. For the other graders, we observe some minor biases between the graders but all of them still share substantial signals and all of the takeaways stay constant under different graders. We refer to the paper for more details.
>
>
> [1] Wang et al. Modifying LLM Beliefs with Synthetic Document Finetuning

---

### Official Review · Reviewer_zV3K · 2025-11-01

**Soundness:** 3
**Presentation:** 4
**Contribution:** 3
**Rating:** 6
**Confidence:** 3

**Summary:**

This paper finds that narrow finetuning of LLMs (a popular technique to study model behavior) creates detectable biases in activation differences between base and finetuned models, which can be used to identify the finetuning objective. The authors apply model diffing techniques (Patchscope, Logit Lens, steering) to analyze a broad range of "model organisms" (narrowly finetuned LLMs) and demonstrate that: (1) activation differences on early tokens of unrelated text encode finetuning domain information, (2) an interpretability agent with access to these diffing techniques that can identify finetuning objectives better than an agent with only black box access, and (3) the narrow / monosemantic nature of the data leads to these strong biases, so mixing pretraining data during finetuning mitigates biases through diversification, albeit also weakening the desired effects of narrow finetuning.

**Strengths:**

1. The paper finds a flaw in current AI safety practice (where narrow finetuning is used extensively), so the findings have practical significance in AI safety research (although the authors do not provide a compelling alternative/solution, as discussed below).
2. The experimental scope is quite comprehensive, covering 33 organisms across 4 families and 7 models from 1B up to 32B parameter scales.
3. The interpretability agent experiment provides an objective automated assessment of how useful / exploitable these activation differences are. These experiments also should be reproducible as the results are reportedly stable across runs.
4. The authors go beyond pure observation of the phenomenon by conducting (well-designed) causal ablation experiments to trace biases back to the semantic homogeneity of the data.

**Weaknesses:**

1. The scope of the paper is quite narrow (no pun intended) because narrow finetuning itself is primarily used as a safety / interpretability tool while in practice training data tends to be more diversified. The authors contend that in more realistic settings (such as domain-adapted vision-language models) the phenomenon is observed to a lesser extent.
2. The paper relies on an interpretability agent based on gpt-5-mini to show that the studied activation biases can be detected easily with the right tools (ADL) while blackbox access to the model is insufficient. The effectiveness of both agents may be highly sensitive to prompt design and it is not clear that the authors tuned both agents equally well.
3. In general, the paper relies heavily on LLM graders. The credibility of the claims would be strengthened by including some human judgments to spot-check whether the graders are calibrated well.
4. The proposed mitigation strategy (mixing in pretraining data) creates a trade-off between strength of activation biases and desired effects of narrow finetuning, and the authors provide no clear guidance on how to best navigate this trade-off.

**Questions:**

1.  What is the reason for using activations from the middle layer? This decision seems quite ad-hoc given that activations vary a lot across layers. Can you observe activation biases in lower or higher layers too?

---

> ### Author Response · Authors · 2025-11-21
>
> We thank the reviewer for their positive feedback and constructive input! We now respond to the points in detail.
>
> **Weaknesses**:
> > 1. Narrow scope.
>
> We would like to highlight that the main takeaway of our paper is that this phenomenon is exactly unique to very narrow finetuning and that it should probably not be used as a proxy for more naturally emerging phenomena. In that sense the point of the paper is exactly that the scope of this phenomenon is narrow. This is very important for the research field because many recent papers have started using very narrowly finetuned models, as the reviewer has highlighted correctly in the strengths. That said, we agree that we haven’t made this point well enough. We have added studies of base<>chat models to show that the traces are not really existing on those finetunes in Appendix E. We have also added a few sentences to make sure that the fact that this is specific to narrow finetuning gets across clearer.
>
> > 2. Reliance on agents and comparison with baselines is very prompt dependent.
> > 3. Heavy reliance on LLM graders.
>
> We agree with the reviewer. Both the ADL agent and the baseline have the exact same prompt except the descriptions of the additional tools/ADL information. We have tried to tune the prompt as well as possible and e.g. make sure that the agent always uses all of its interaction budget. We have further added extensive analysis with respect to the effect of different graders in Section 4.1 as well as Appendix D. This includes:
> - Running the full agent also with Gemini 2.5 Pro additionally to GPT-5 (medium)
> - Extending the hypothesis grading to graders Gemini 2.5 Flash and Claude Haiku 4.5 including 3 reruns per agent hypothesis and grader model.
> - Running the token relevance grading and patchscope scaling factor grading additionally with Gemini 2.5 Lite and Claude Haiku 4.5.
> - Running the steering factor additionally with Gemini 2.5 Flash Lite
>
> We evaluate a) whether these graders result in systematic biases using hierarchical generalized linear models (GLM) within the HiBayes framework (https://arxiv.org/pdf/2505.05602, https://arxiv.org/html/2507.03772v2) and b) whether the graders agree through correlation analysis and Krippendorff’s alpha. Results show that the agent model has no systematic impact on the grade and all hypothesis graders agree to a very high degree. For the other graders, we observe some minor biases between the graders but all of them still share substantial signals and all of the takeaways stay constant under different graders. We refer to the paper for more details. We hope that this helps alleviate the reviewers' concern.
>
> > 4. No clear guidance for how to deal with the mixing strength and the desired effects.
>
> This is a fair point. There is a clear pareto frontier of mixing data vs. capabilities and we think it is important for future work to understand what kind of data is best to put in the mix and how you can alleviate the traces best. We think this is out of scope for the current paper. Nonetheless, we have added a sentence that “we recommend that practitioners mix in as much unrelated data as possible when training model organisms, while ensuring the model still retains the initial finetuning objective.”
>
> **Questions**:
>
> > 1. Why middle layer?
>
> This choice builds on a range of literature that shows that middle layers are where we expect to find the richest and most interesting representations [1,2] (we have added a reference to those). There’s a range of interpretability literature that has focused on the middle literature because of this (e.g. [2,3]).
>
> [1]: Skeal et al.: Layer by Layer: Uncovering Hidden Representations in Language Models
>
> [2]: Ali et al.: Entropy-Lens: The Information Signature of Transformer Computations
>
> [2]: Lindsey et al.: Sparse Crosscoders for Cross-Layer Features and Model Diffing
>
> [3]: Minder et al.: Overcoming Sparsity Artifacts in Crosscoders to Interpret Chat-Tuning

---

### Official Review · Reviewer_YvCS · 2025-11-05

**Soundness:** 3
**Presentation:** 3
**Contribution:** 3
**Rating:** 8
**Confidence:** 4

**Summary:**

The paper introduces a model diffing framework, the Activation Difference Lens (ADL), designed to extract clearly readable traces of LLMs' internalized objectives resulting from narrow finetuning. The core methodology involves measuring the differences between the activations (δ) of a base model and a narrowly finetuned model on the first few tokens of random pretraining corpus data. Their methodology is clearly explained, leveraging two key interpretability tools applied to the activation differences:  Logit Lens, Patchscope (modified), which reveals relevant tokens), and steering (which amplifies the model’s tendency to generate text highly similar to the finetuning data). The paper develops an LLM-based interpretability agent which uses these ADL results and LLM graders to form and verify hypotheses about the finetuning objective.

**Strengths:**

The main contribution is the demonstration that these few-token activation differences encode salient traces of the training objective. The empirical study, spanning a wide range of model organisms and architectures, proves the ADL framework is highly effective compared to existing blackbox baselines using only simple prompting. Through causal ablation analysis and loss calculation, the paper makes a sound argument that the easily detectable biases highlighted by ADL are somewhat tied to overfitting to the semantically homogeneous finetuning data. Finally, through mitigation experiments, the authors proposed an effective method to avoid these strong biases by mixing unrelated pretraining data into the narrow finetuning datasets, which substantially reduces the detectable bias. The paper also highlights a warning to the interpretability and AI safety communities. These findings suggest finetuning signals may be artificially strong and overpower traces from standard broader finetuning. Therefore, using such model organisms as proxies for realistic scenarios may compromise their validity.

**Weaknesses:**

In general, this study brings important insights to narrow finetune and model organism study. However, some caveats and additional experiments could make even sounder arguments.

- The empirical study involves extensive usage of LLM grader. Although it is mentioned as a limitation, It would be nice to see some expansion on generalization of such a framework with other LLM graders other than gpt-5-mini.
- The causal effect analysis is only done on three models, relatively small sized. Expanding the study to a wider range of models architecture and of model sizes will help make a stronger argument of overfitting. Further, the inconsistent result from Gemma makes the conclusion harder to believe. Experiments on the older Gemma model could help to further ground the hypothesis for its inconsistent behavior.

**Questions:**

- In section 4.2, it is discussed that similar activation difference analysis was also done between the base model and the narrowly finetuned model, where bias detectable similar to that of chat model and finetuned model. Although the argument is sound, it seems the setup overlooks the potential impact of activation difference between base model and chat model. Can a similar analysis be done between the base model and the chat model? Would that help control for confounders?

---

> ### Author Response · Authors · 2025-11-21
>
> We thank the reviewer for their very positive feedback and constructive input! We now respond to the points in detail.
>
> **Weaknesses**:
> > The empirical study involves extensive usage of LLM grader. Although it is mentioned as a limitation, It would be nice to see some expansion on generalization of such a framework with other LLM graders other than gpt-5-mini.
>
> We agree with the reviewer and have added extensive analysis with respect to the effect of different graders in Section 4.1 as well as Appendix D. This includes:
> - Running the full agent also with Gemini 2.5 Pro additionally to GPT-5 (medium)
> - Extending the hypothesis grading to graders Gemini 2.5 Flash and Claude Haiku 4.5 including 3 reruns per agent hypothesis and grader model.
> - Running the token relevance grading and patchscope scaling factor grading additionally with Gemini 2.5 Lite and Claude Haiku 4.5.
> - Running the steering factor additionally with Gemini 2.5 Flash Lite
>
>
> We evaluate a) whether these graders result in systematic biases using hierarchical generalized linear models (GLM) within the HiBayes framework (https://arxiv.org/pdf/2505.05602, https://arxiv.org/html/2507.03772v2) and b) whether the graders agree through correlation analysis and Krippendorff’s alpha. Results show that the agent model has no systematic impact on the grade and all hypothesis graders agree to a very high degree. For the other graders, we observe some minor biases between the graders but all of them still share substantial signals and all of the takeaways stay constant under different graders. We refer to the paper for more details. We hope that this helps alleviate the reviewers' concern.
>
> > The causal effect analysis is only done on three models, relatively small sized. Expanding the study to a wider range of models architecture and of model sizes will help make a stronger argument of overfitting. Further, the inconsistent result from Gemma makes the conclusion harder to believe. Experiments on the older Gemma model could help to further ground the hypothesis for its inconsistent behavior.
>
> We agree with the reviewer but would like to highlight that the results do actually hold up for gemma: On the finetuning dataset the intervention results in a spike in loss. The only thing that is inconsistent with gemma is that the loss also increases on the pretraining dataset, which we hypothesize is because “this model changed sufficiently during finetuning that the ablated directions became crucial for the finetuned model's computation, making replacement with base model activations generally harmful.” (L456). We also would like to highlight that we do this analysis for all SDF organisms for the three models, so 15 organisms across three models in total.
>
> **Questions**:
> > In section 4.2, it is discussed that similar activation difference analysis was also done between the base model and the narrowly finetuned model, where bias detectable similar to that of chat model and finetuned model. Although the argument is sound, it seems the setup overlooks the potential impact of activation difference between base model and chat model. Can a similar analysis be done between the base model and the chat model? Would that help control for confounders?
>
> No, this phenomenon is specific to finetunes with semantically very narrow datasets. We have made sure to emphasize this argument better. We have also added new base<>chat experiments that show that the traces are not really existing on those finetunes in Appendix E.

---

### Author Response · Authors · 2025-11-21

We would like to thank all the reviewers for their feedback and very positive responses. We have tried our very best to address all the points raised; most importantly, we have conducted a comprehensive analysis of all the stochastic LLM graders. Please see the individual responses for details.

We have updated the PDF and marked all the major changes in purple.

---

### Meta-Review · Area_Chair_z1uG · 2026-01-03

**Summary:**

Reviewers expressed concerns about heavy reliance on LLM graders without testing multiple models or analyzing biases, limited scope and inconsistency in causal effect analyses across few small models (especially Gemma), prompt sensitivity in agent comparisons, lack of guidance on data mixing trade-offs, unclear motivation for middle-layer focus, and the derivative nature of the methodology combining existing tools, informing a suggested acceptance due to strong empirical findings, novelty in implications for safety/interpretability, and thorough rebuttal but with calls for broader generalization.

**Reviewer Concerns:**

The rebuttal addressed grader reliance through multi-model analyses showing no systematic biases and high agreement, causal inconsistencies by clarifying Gemma results and adding base-chat experiments, middle-layer choice with added references, abstract edits mentioning ADL, and mixing guidance via a new recommendation; outstanding concerns include expanding causal analyses to more/larger models, incorporating human judgments for calibration, detailed prompt tuning equality, and computational costs of mixing.

**Reviewer Scores:**

All reviewers may still keep the same scores as they have accepted the positive parts of the paper.

---

### Decision · Program_Chairs · 2026-01-26

Accept (Poster)